## META-RESEARCH

# A comprehensive review of randomized clinical trials in three medical journals reveals 396 medical reversals

**Abstract** The ability to identify medical reversals and other low-value medical practices is an essential prerequisite for efforts to reduce spending on such practices. Through an analysis of more than 3000 randomized controlled trials (RCTs) published in three leading medical journals (the Journal of the American Medical Association, the Lancet, and the New England Journal of Medicine), we have identified 396 medical reversals. Most of the studies (92%) were conducted on populations in high-income countries, cardiovascular disease was the most common medical category (20%), and medication was the most common type of intervention (33%).
DOI: https://doi.org/10.7554/eLife.45183.001

**DIANA HERRERA-PEREZ[†], ALYSON HASLAM[†], TYLER CRAIN[†], JENNIFER GILL[†], CATHERINE LIVINGSTON, VICTORIA KAESTNER, MICHAEL HAYES, DAN MORGAN, ADAM S CIFU AND VINAY PRASAD***

## Introduction

Low-value medical practices are medical practices that are either ineffective or that cost more than other options but only offer similar effectiveness (*Prasad et al., 2013*; *Prasad et al., 2011*; *Schpero, 2014*). Such practices can result in physical and emotional harm, undermine public trust in medicine, and have both an opportunity cost (*Korenstein et al., 2018*) and a financial cost (*Reid et al., 2016*; *Beaudin-Seiler, 2016*). Identifying and eliminating low-value medical practices will, therefore, reduce costs and improve care.

Medical reversals are a subset of low-value medical practices and are defined as practices that have been found, through randomized controlled trials, to be no better than a prior or lesser standard of care (*Prasad et al., 2013*; *Prasad et al., 2011*). It can, however, be difficult to identify medical reversals. For example, Cochrane reviews provide high-quality evidence on medical practices (*Garner et al., 2013*), but each review focuses on only one practice and

many practices have not been reviewed by Cochrane. The Choosing Wisely initiative in the US maintains a list of low-value medical practices, but it relies on medical organizations to report such practices and often includes only those practices where there is a high degree of consensus (*Beaudin-Seiler, 2016*).

Here we report how a systematic search of randomized controlled trials in three leading medical journals – the Journal of the American Medical Association (JAMA), the Lancet, and the New England Journal of Medicine (NEJM) – identified 396 medical reversals. It is our hope that, by building on previous efforts in this area (*Prasad et al., 2013*), this list will help others to eliminate the use of these practices.

## Results

We reviewed JAMA and the Lancet between 2003 and 2017, and NEJM between 2011 and 2017, and identified a total of 7036 original articles (*Figure 1*; 2911 in JAMA, 2624 in the

**\*For correspondence:** prasad@ohsu.edu

[†]These authors contributed equally to this work

Lancet, and 1501 in NEJM). There were 3017 articles reporting the results of randomized control trials regarding a medical practice, and these articles were further coded for novelty/establishment and whether the outcomes were positive, negative, or inconclusive. After excluding studies that were novel (n = 1373) or established with positive or inconclusive outcomes (n = 1229), there were 415 (14%) studies identified as tentative medical reversals. After a search of systematic reviews to refute these tentative reversals, 19 were excluded, leaving a total of 396 medical reversals (6% of all original articles and 13% of all randomized trials).

Many of these 396 reversals had been the subject of systematic reviews: in 209 cases (53%) the systematic review confirmed that the medical practice in question was indeed a medical reversal; in 109 cases (28%) the results of the systematic review were inconclusive; and for 78 cases (20%) there was no systematic review. 154 of the reversals (39%) were found in JAMA, 129 (33%) were found in NEJM, and 113 (29%) were found in Lancet.

Reversal study characteristics are described in *Table 1*. Most studies (92%, n = 366) were conducted on populations in high-income countries, whereas 8% (n = 30) were done in low- or middle-income countries, including, but not limited to China, India, Malaysia, Ghana Tanzania, and Ethiopia. Cardiovascular disease was the most common medical category (20%, n = 80), followed by public health/preventive medicine (12%, n = 48), and critical care (11%, n = 45). Regarding the type of intervention, medication was the most common (33%, n = 129), followed by a procedure (20%, n = 81), vitamin/supplement (13%, n = 53), device (9%, n = 35) and system intervention (8%, n = 30). The breakdown of funding categories were as such (*Supplementary file 1*): 253 (63.9%) were from non-industry sources only; 88 (22.2%) were from a combination of industry and non-industry sources; 36 (9.1%) from industry only sources; and 3 (0.8%) from non-industry sources plus insurance company (n = 2) or a development bank (n = 1). There were 16 (4.0%) studies that we could not find the source of funding.

*Table 2* summarizes 20 selected medical reversals. The selected examples were chosen to represent various types of practices in various medical disciplines over the full years that we did the analysis. *Supplementary file 2* contains a full list of reversal summaries. *Figure 2* shows the percent of articles that are in each journal, by medical specialty.

## Discussion

Here we present a broad and extensive list of established medical practices found to be ineffective in randomized control trials. This list represents practices from all disciplines of medical care. These practices add to a previously reported list of 146 medical reversals published during years 2001–2010 (*Prasad et al., 2013*).

Efforts to identify low-value practices are numerous. In the US Choosing Wisely initiative began by asking members of each medical specialty to provide a list of the top five diagnostic tests or treatments that are expensive and have evidence showing a lack of benefit (*Schpero, 2014*): similar initiatives have been implemented in other countries (*de Vries et al., 2016*). Some have performed systematic searches of the scientific databases using key words (*de Vries et al., 2016*). Others have used a multiplatform attempt, consisting of searching the peer-reviewed literature, insurance and health organization databases, and opportunistic samplings of knowledgeable experts in the field (*Elshaug et al., 2012*). Each of these ways to identify medical reversals or low-value practices has advantages and disadvantages, but identifying these practices can be challenging because of their heterogeneity, the lack of established methods to identify these practices, the difficulty in applying them to the correct population or subpopulation, and the obstacle of prioritizing which practices are more or less low-value (*Elshaug et al., 2013*).

Prior work by Schwartz and colleagues approximated the financial costs of 26 low-value services that are more commonly used in the older adult population (*Schwartz et al., 2014*). They estimated that spending for these services in the Medicare population was between $1.9 and $8.5 billion during 2008–2009, which was between 0.6% and 2.7% of Medicare Parts A and B spending. In their analysis, at least 25% of Medicare beneficiaries received low-value services during 2008–2009. These results are especially notable considering the authors only used the 26 most commonly used low-value services. In contrast, the ubiquity of medical reversals has been previously reported upon in the NEJM, where 146 practices were identified as medical

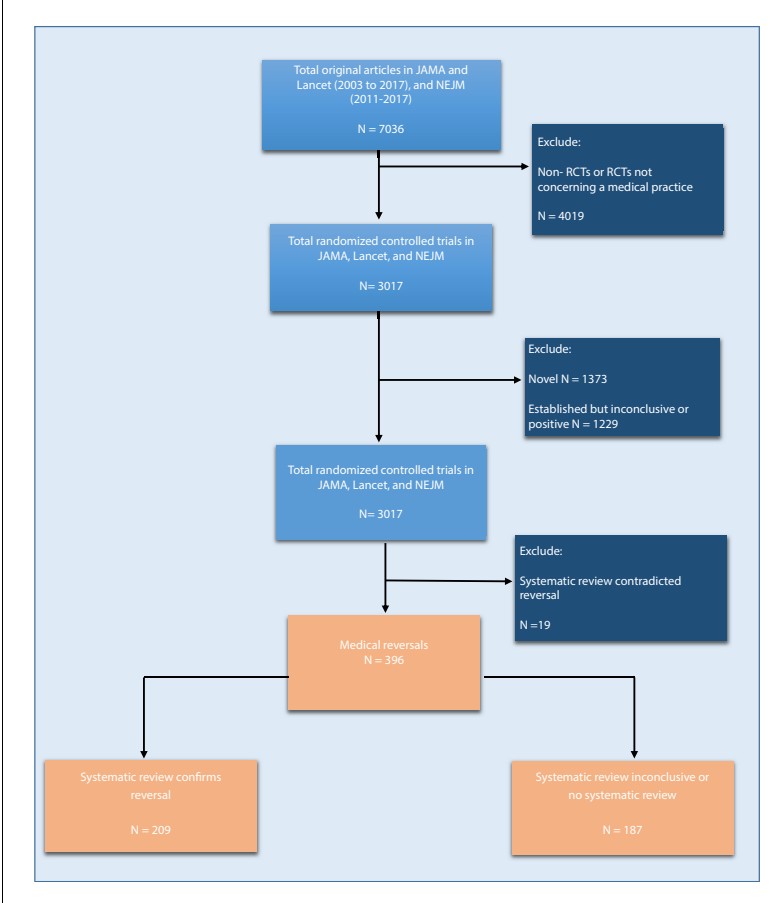

**Figure 1.** Flowchart of selection process to identify included randomized trials.
DOI: https://doi.org/10.7554/eLife.45183.002

reversals over a decade (**Prasad et al., 2013**). Here, we hope to add to the prior efforts of others in providing a larger and more comprehensive list (396 practices in total) for clinicians and researchers to guide practice as they care for patients more effectively and more economically.

We found reversals in a variety of medical sub-fields and types of devices, procedures, or practices. These reversals had been practiced and tested in high-income as well as low- to middle-income countries, although the highest percentage of reversals was in high-income countries, likely because most randomized trials are performed in this setting. In countries like the US, where there was a 20% increase in spending between 2013 and 2015, and drug prices alone surpassed the increase in aggregate health care spending (**Kesselheim et al., 2016**), the identification and disuse of costly and ineffective (or possibly harmful) medications and practices are especially important. For example,

bevacizumab (Avastin) was approved in 2008 by the Food and Drug Administration (FDA) in the US for metastatic breast cancer under the accelerated approval program, but was later shown to not improve overall survival (**Vitry et al., 2015**), even though the cost to each patient was $88,000 per year (**Selyukh, 2011**). Consequently, the FDA approval for that indication was withdrawn in November 2011 (**Vitry et al., 2015**).

Reversals were not just limited to practices performed by physician or health care providers only. Many reversals involved practices where the physician was a 'gatekeeper' to access these practices, but some were practices where the patient could access on their own, such as behavioral practices (e.g., cognitive behavioral therapy or mindfulness interventions), complementary or non-traditional practices (e.g., acupuncture), dietary supplements (e.g. omega-3 fatty acids or vitamin A supplementation), community practices (e.g., programs to prevent teenage pregnancy or self-poisoning), or wearable technology. Wearable technology has become especially popular among people who are interested in tracking their physical activity in an effort to lose weight. A study on the use of wearable technology, however, found that weight loss was significantly less among the group that had access to wearable technology, compared to the group that did not (**Jakicic et al., 2016**). With increasing availability of healthcare interventions that are readily accessible to everyone without a prescription, there needs to be greater discussion on whether these work between patients and physicians, as well as discussion on the regulation of these interventions.

13% of all randomized trials were medical reversals: this is slightly higher than a previous report based on an analysis of just one journal. There was some variation in the percentage of trials published in each journal that reported on practices considered as a medical reversal, ranging from 29% (113/396) for Lancet to 39% (154/396) for JAMA.

Finally, reversals highlight the importance of independent, governmental and non-conflicted funding of clinical research. The majority of reversal studies we found were funded by such sources (63.9%), with a minority funded solely by the industry (9.1%). Conversely, industry funded research represented between 35–49% of trials registered on ClinicalTrials.gov during years 2006 through 2014 (**Ehrhardt et al., 2015**).

**Table 1.** Characteristics of the included reversal studies from JAMA (2003–2017), Lancet (2003–2017), NEJM (2011–2017).

| | JAMA (n = 154) | Lancet (n = 113) | NEJM (n = 129) | Total (N = 396) |
|---|---|---|---|---|
| **Economic status of studied population** | | | | |
| High-income | 152 (99%) | 93 (82%) | 121 (94%) | 366 (92%) |
| Low- and middle-income | 2 (1%) | 20 (18%) | 8 (6%) | 30 (8%) |
| **Intervention type** | | | | |
| Medication | 49 (32%) | 36 (32%) | 44 (34%) | 129 (33%) |
| Procedure | 22 (14%) | 22 (19%) | 37 (29%) | 81 (20%) |
| Vitamins/supplements/food | 34 (22%) | 11 (11%) | 8 (6%) | 53 (13%) |
| Device | 15 (10%) | 12 (11%) | 8 (6%) | 35 (9%) |
| System intervention | 13 (8%) | 12 (11%) | 5 (4%) | 30 (8%) |
| Optimize | 5 (3%) | 5 (4%) | 13 (10%) | 23 (6%) |
| Behavioral therapy | 6 (4%) | 4 (4%) | 2 (2%) | 12 (3%) |
| Screening test | 3 (2%) | 4 (4%) | 2 (2%) | 9 (2%) |
| Treatment algorithm | 1 (1%) | 3 (3%) | 5 (4%) | 9 (2%) |
| Diagnostic instrument | 2 (1%) | 2 (1%) | 2 (2%) | 6 (2%) |
| Radiation | 2 (1%) | 1 (1%) | 2 (2%) | 5 (1%) |
| Over-the-counter medication | 2 (1%) | 1 (1%) | 1 (1%) | 4 (1%) |
| | | | | |
| **Medical Discipline** | | | | |
| Cardiovascular disease | 21 (14%) | 15 (13%) | 44 (34%) | 80 (20%) |
| Public health/preventive medicine | 32 (20%) | 13 (12%) | 3 (2%) | 48 (12%) |
| Critical care medicine | 18 (12%) | 6 (5%) | 21 (16%) | 45 (11%) |
| Obstetrics and gynecology | 13 (8%) | 13 (12%) | 10 (8%) | 36 (9%) |
| Neurology/neurosurgery | 7 (5%) | 10 (9%) | 8 (6%) | 25 (6%) |
| Oncology | 7 (5%) | 12 (11%) | 4 (3%) | 23 (6%) |
| Orthopedic | 15 (10%) | 5 (4%) | 3 (2%) | 23 (6%) |
| Pulmonary disease | 6 (4%) | 11 (10%) | 5 (4%) | 22 (6%) |
| Pediatrics | 2 (1%) | 6 (5%) | 7 (5%) | 15 (4%) |
| Gastroenterology/hepatology | 6 (4%) | 3 (4%) | 4 (3%) | 13 (3%) |
| Endocrinology, diabetes, metabolism | 7 (5%) | 0 (0%) | 5 (4%) | 12 (3%) |
| Psychiatry | 4 (3%) | 5 (4%) | 1 (1%) | 10 (3%) |
| Nephrology | 4 (3%) | 4 (4%) | 2 (2%) | 10 (3%) |
| Infectious disease | 2 (1%) | 3 (3%) | 3 (2%) | 8 (2%) |
| Surgery | 2 (1%) | 2 (2%) | 4 (3%) | 8 (2%) |
| Urology | 3 (2%) | 3 (3%) | 1 (1%) | 7 (2%) |
| Allergy and immunology | 1 (1%) | 0 (0%) | 2 (2%) | 3 (1%) |
| Anesthesiology | 1 (1%) | 2 (2%) | 1 (1%) | 4 (1%) |
| Rheumatology | 2 (1%) | 0 (0%) | 1 (1%) | 3 (1%) |
| Ophthalmology | 1 (1%) | 0 (0%) | 0 (0%) | 1 (<1%) |

Column percentage may not add up to 100% because of rounding.

DOI: https://doi.org/10.7554/eLife.45183.003

### Strengths and limitations

There are several strengths and limitations to this paper. First, we looked at just three journals (each of which has a high impact factor). Results may not be broadly generalizable to all journals or fields, and reversals in our list could be affected by the editors' decision to publish or not publish a given article. Second, documented

**Table 2.** Selected reversal summaries from JAMA (2003–2017), Lancet (2003–2017), NEJM (2011–2017).

| RCT and medical discipline | Reversal summary | Systematic review conclusion |
|---|---|---|
| Morris et al. 2016. Immediate delivery compared with expectant management after preterm pre-labour rupture of the membranes close to term (PPROMT trial): a randomized controlled trial. *The Lancet* 387:444–452. (1/30/2016) [Obstetrics and gynecology] | Both the American College of Obstetricians and Gynecologists and Royal College of Obstetrics and Gynaecology support and/or recommend immediate delivery for women with ruptured membranes who are 34 weeks or greater (*Morris et al., 2016*). Neonatal infection is a major concern in when there has been a ruptured membrane, especially in premature infants (*Merenstein and Weisman, 1996*). In this trial, participants assigned to the expectant management group did not have any worse outcomes regarding the primary outcomes of neonatal sepsis (2%; n = 924 in the immediate birth arm vs. 3%; n = 915 in the expectant management arm; RR = 0.8; 95% CI = 0.5–1.3; p=0.37) or neonatal morbidity and mortality (8% vs. 7%; p=0.32) than those assigned to immediate delivery, and had less respiratory distress (p=0.008) and need for mechanical ventilation (p=0.02). This is a reversal of the practice of immediate delivery in women with preterm, pre-labor rupture of the membranes, as it does not lead to less neonatal sepsis. | 2017. Cochrane review. "We found no clinically important difference in the incidence of neonatal sepsis between women who birth immediately and those managed expectantly in PPROM prior to 37 weeks' gestation. Early planned birth was associated with an increase in the incidence of neonatal RDS, need for ventilation, neonatal mortality, endometritis, admission to neonatal intensive care, and the likelihood of birth by caesarean section, but a decreased incidence of chorioamnionitis." (*Bond et al., 2017*) |
| Edmond et al. 2015. Effect of early neonatal vitamin A supplementation on mortality during infancy in Ghana (Neovita): a randomised, double-blind, placebo-controlled trial. *The Lancet* 385:1315–1323. (4/4/2015) [Pediatrics] | Vitamin A deficiency is a public health issue in low-income countries. While multiple trials, including a Cochrane review, have been performed on the effectiveness of vitamin A supplementation in infants in low-income countries, the WHO stated that there was insufficient evidence to make a recommendation on its usage (*Gogia and Sachdev, 2011*; *Imdad et al., 2016*). The International Vitamin A Consultative Group (IVACG) supports the use of 50,000 IUs for infants < 6 months of age (*Ross, 2002*). In this trial based in Ghana, vitamin A supplementation did not lead to a lower mortality rate compared to placebo (24.5/1,000 [n = 11,474] vs. 21.8/1,000 [n = 11,481] supplemented infants; RR1.12; 95% CI = 0.95–1.33; p=0.18), in newborn infants. This is a reversal of the practice of vitamin A supplementation during the early neonatal period in Africa, as it does not improve mortality. | 2017. Cochrane review. "Evidence provided in this review does not indicate a potential beneficial effect of vitamin A supplementation among neonates at birth in reducing mortality during the first six months or 12 months of life." (*Haider and Bhutta, 2017*) |
| Conjee et al. 2011. Sertraline or mirtazapine for depression in dementia (HTA-SADD): a randomised, multicentre, double-blind, placebo-controlled trial. *The Lancet* 378:403–411. (7/30/2011) [Psychiatry] | Sertraline and mirtazapine are commonly prescribed for depression in older adults, and mirtazapine is recommended as a first-line treatment for depression in clinical guidelines, regardless of age (*Nelson et al., 2008*; *Doody et al., 2001*; *Eccles et al., 1998*; *National Collaborating Centre for Mental Health, 2007*). The results from this trial show that neither sertraline (n = 107; mean difference = 1.17; 95% CI = −0.23 to 2.58; p=0.10) nor mirtazapine (n = 108; mean difference = 0.01; 95% CI = −1.37 to 1.38; p=0.99) improved rates of depression over placebo (n = 111) in those with Alzheimer's disease. This is a reversal of the practice of using traditional treatments for depression, such as sertraline or mirtazapine, in patients with Alzheimer's, as depression in this population may have different mechanisms than that of the general population. | 2017. "We found no significant drug-placebo difference for depressive symptoms. Overall quality of the evidence was moderate because of methodological limitations in studies and the small number of trials." (*Orgeta et al., 2017*) |

*Table 2 continued on next page*

*Table 2 continued*

| RCT and medical discipline | Reversal summary | Systematic review conclusion |
|---|---|---|
| Dennis et al. 2009. Effectiveness of thigh-length graduated compression stockings to reduce the risk of deep vein thrombosis after stroke (CLOTS trial 1): a multicentre, randomised controlled trial. *The Lancet* **373**:1958–1965. (6/9/2009) [Cardiovascular] | Compression therapy was first used by German physicians in the late 19th century when they noticed that superficial vein thromboses disappeared after the use of compression bandages (*Galanaud et al., 2013*). Compression stockings were used as early as the 1930 s but became widely used after the results of a trial were published in 2000 (*Galanaud et al., 2013*). National stroke guidelines recommend use of graduated compression stockings (GCS) to reduce risk of deep vein thrombosis (DVT) and pulmonary embolism (*Adams et al., 2007*) although there is a lack of clinical trials investigating its use in an acute stroke population. This study compared routine care plus GCS (n = 1265) with routine care plus avoidance of GCS (n = 1262) in patients within 1 week of an acute stroke. The study found that there was no difference in occurrence of symptomatic or asymptomatic DVT between groups (126 [10%] in the GCS group vs 133 [10.5%] in the control group) and more adverse events (64 [5%] vs 16 [1%]) in the GCS group. This is a reversal of the use of thigh-length graduated compression stockings to reduce the risk of deep vein thrombosis after stroke. | 2010. Cochrane review. 'Evidence from randomised trials does not support the routine use of GCS to reduce the risk of DVT after acute stroke." (*Naccarato et al., 2010*) However, this RCT was not included in the review. |
| Moss et al. 2006. Effect of mammographic screening from age 40 years on breast cancer mortality at 10 years' follow-up: a randomized controlled trial. *The Lancet* **368**:2053–2060. (12/9/2006) [Public health and general preventive medicine] | In the past, the American Cancer Society recommended that women between the ages of 40 and 49 get mammograms every 1–2 years (*American Cancer Society, 2018*). However, the benefit of mammograms for women under the age of 50 has not been established. 160 921 women aged 39–41 years old were randomly assigned in the ratio of 1:2 to an intervention group of annual mammography to age 48 or to a control group of usual medical care. At a mean follow-up of 10.7 years, there was no significant difference in breast cancer mortality between the intervention and control groups (relative risk 0.83 [95% CI 0.66–1.04], p=0.11). This is a reversal of the recommendation of mammographic screening every 1–2 years for women ages 40–49. | 2013. Cochrane review. "The chance that a woman will benefit from attending screening is small at best, and - if based on the randomised trials - ten times smaller than the risk that she may experience serious harm in terms of overdiagnosis." (*Gøtzsche and Jørgensen, 2013*) |
| Kerr et al. 2003. Intrahepatic arterial versus intravenous fluorouracil and folinic acid for colorectal cancer liver metastases: a multicentre randomised trial. *The Lancet* **361**:368–373. (2/1/2003) [Oncology] | Colon cancer, one of the most common types of cancer, has a relapse rate, after surgery, of about 50%, with the liver being a common site for metastasis (*Midgley and Kerr, 1999*). Intrahepatic arterial infusion has been used as a method of delivering chemotherapy because it is thought that there would be a higher dose of chemotherapy to cancer cells, while lessoning the side-effects of chemotherapy (*Ansfield et al., 1971*; *Fortner et al., 1984*). This trial randomly allocated 290 patients from 16 centers to receive either intravenous chemotherapy (folinic acid 200 mg/m$^2$, fluorouracil bolus 400 mg2 and 22 hr infusion 600 mg/m$^2$, day 1 and 2, repeated every 14 days) or IHA chemotherapy designed to be equitoxic (folinic acid 200 mg/m$^2$, fluorouracil 400 mg/m$^2$ over 15 mins and 22 hr infusion 1600 mg/m$^2$, day 1 and 2, repeated every 14 days). Median survival in the IHA group was 14.7 months and was 14.8 months in the intravenous group (hazard ratio 1.04 [95% CI 0.80–1.33]). This is a reversal of the use of IHA for patients with colorectal cancer liver metastases. | 2011. Cochrane review. "Currently available evidence does not support the clinical or investigational use of fluoropyrimidine-based HAI alone f or the treatment of patients with unresectable CRC liver metastases: in fact, the greater tumor response rate obtained with this HAI regimen does not translate into a survival advantage over fluoropyrimidine alone SCT." (*Mocellin et al., 2009*) |

*Table 2 continued on next page*

*Table 2 continued*

| RCT and medical discipline | Reversal summary | Systematic review conclusion |
|---|---|---|
| MUST trial group. 2017.<br>Association Between Long-Lasting Intravitreous Fluocinolone Acetonide Implant vs Systemic Anti-inflammatory Therapy and Visual Acuity at 7 Years Among Patients With Intermediate, Posterior, or Panuveitis. *JAMA* **317**:1993–2005. (5/16/2017) [Ophthalmology] | Noninfectious intraocular inflammation, or uveitis, can lead to visual impairment. Currently, there are two treatments commonly used for uveitis; the first approach is through systemic corticosteroids and corticosteroid-sparing immunosuppressive drugs (*Jabs et al., 2005*). The other, more recent approach was approved by the FDA in 2005 and involves surgically implanting fluocinolone acetonide implants (*Callanan et al., 2008*). When systemic therapy (n = 126) and intravitreous implants (n = 129) approaches were compared with one another in a randomized control trial, it was found that after seven years of follow up, those that were randomized to receive implants had poorer visual acuity than the group who were treated with systemic therapy. Change in mean visual acuity from baseline through 7 years was 1.15 in the systemic therapy group and −5.96 in the implant group (between-group difference, −7.12; 95% CI, −12.4 to −2.14; p=0.006). This is a reversal of intravitreous fluocinolone acetonide implants for uveitis. | None found |
| Jakicic et al. 2016. Effect of Wearable Technology Combined With a Lifestyle Intervention on Long-term Weight Loss The IDEA Randomized Clinical Trial. *JAMA* **316**:1161–1171. (9/20/2016) [Public health and general preventive medicine] | Wearable technologies have become increasingly popular as tools to assist in weight loss since they can help track physical activity and estimate calorie burn (*Piwek et al., 2016*). This clinical trial randomized adults who were participating in a weight-loss program (including a low-calorie diet, increases in physical activity, group counseling sessions, telephone counseling sessions, text message prompts, and access to study materials on a website) to use a wearable device and accompanying web interface (enhanced intervention group, n = 237) or to a self-monitoring website (standard intervention group, n = 233). The study found that the standard intervention group experienced significantly more weight loss than the enhanced intervention group after 24 months (5.9 kg vs 3.5 kg; difference 2.4 kg; 95% CI, 1.0–3.7; p=0.002). This is a reversal of wearable technology for long-term weight loss. | 2017. While this review concluded that wearable technology reduces sedentary behavior, there were no SR/MA on whether these devices reduce weight (*Stephenson et al., 2017*). This review did not include the RCT. |
| Manson et al. 2013. Menopausal Hormone Therapy and Health Outcomes During the Intervention and Extended Poststopping Phases of the Women's Health Initiative Randomized Trials<br>*JAMA* **310**:1353–1368. (10/2/2013) [Obstetrics and gynecology] | Postmenopausal hormone replacement therapy (HRT) was initially used in the 1940 s as a way to delay age-related health outcomes, but in the 1970's studies began to emerge showing that the use of HRT, specifically unopposed estrogen, was associated with endometrial cancer. Progesterone was thought to oppose the effects of estrogen and mitigate the excess risk of cancer, so women began to take them again. By the 1990s, HRTs were the most commonly prescribed medications (*Brett and Madans, 1997*). The Women's Health Initiative investigated the effects of HRT in postmenopausal women compared to placebo. This paper is an overview of the many health effects of HRT and found that there is a complex pattern of risks and benefits. The authors concluded that HRT is not an appropriate or recommended intervention for the prevention of chronic disease in postmenopausal women. | 2015. "The current evidence suggests that MHT [menopausal hormone therapy] does not affect the risk of death from all causes, cardiac death and death from stroke or cancer." (*Benkhadra et al., 2015*) Another SR/MA (2016) did not find any cardiovascular benefit to hormone therapy (*Mahmoodi et al., 2017*). |

*Table 2 continued on next page*

*Table 2 continued*

| RCT and medical discipline | Reversal summary | Systematic review conclusion |
| --- | --- | --- |
| Siversten et al. 2006. Cognitive Behavioral Therapy vs Zopiclone for Treatment of Chronic Primary Insomnia in Older Adults A Randomized Controlled Trial. *JAMA* **295**:2851–2858. (6/28/2006) [Public health and general preventive medicine] | Insomnia is a common complaint among individuals age 55 years and older and is associated with reduced quality of life, affective disorders, and increased health service utilization (**Simon and VonKorff, 1997**). Pharmacological interventions are common treatments prescribed by primary care physicians, yet sleep medication has shown to have a small effect size and clinical benefit, and long-term use of the drugs can cause dependency and increased tolerance (**Glass et al., 2005**). Zopiclone, a non-benzodiazepine sleeping pill, is also associated with next-day sleepiness and traffic collisions (**Allain et al., 1991**; **Montplaisir et al., 2003**). Cognitive behavioral therapy (CBT) is the most widely used psychological intervention for insomnia but has limited studies proving its efficacy. This study was the first RCT to compare the effects of nonbenzodiazepine sleep medications with nonpharmacological treatment. The study found that, at 6 months, CBT improved sleep efficiency from 81.4% to 90.1% compared to the zopiclone group, which saw a decrease in efficiency from 82.3% to 81.9%. CBT (n = 18) improved short and long-term sleep outcomes compared to zopiclone and that in most outcomes, zopiclone (n = 16) was no more effective than placebo (n = 12). Zopiclone was no better than placebo in improving symptoms for patients with insomnia. This is a reversal of zopiclone for improving insomnia symptoms. | 2012. "There is moderate grade evidence suggesting CBT-I is superior to the non-benzodiazepines zopiclone and zolpidem for improving sleep measures in the short term." (**Mitchell et al., 2012**) |
| Hallstrom et al. 2006. Manual Chest Compression vs Use of an Automated Chest Compression Device During Resuscitation Following Out-of-Hospital Cardiac Arrest: A Randomized Trial. *JAMA* **295**:2620–2628. (6/14/2006) [Cardiovascular] | Out-of-hospital cardiac arrest is generally treated by cardiopulmonary resuscitation (CPR) and the quality and order of resuscitation intervention may have an effect on cardiac and neurological outcomes (**Steen et al., 2003**). Consistent compressions in CPR is difficult while maintaining quality, and paramedics have been shown to provide shallower, slower compressions over time (**Ochoa et al., 1998**). Manual chest compression devices were designed to provide ideal chest compressions. The AutoPulse Resuscitation System is a load-distributing band circumferential chest compression device (LDB-CPR) that received marketing clearance by the FDA in 2002 (**Food and Drug Administration, 2019**). This study compared the use of an LDB-CPR device with manual CPR in EMS care for patients with cardiac arrest that was presumed to be of cardiac origin and that had occurred prior to the arrival of EMS personnel. Automated LDB-CPR devices (n = 394) were associated with worse neurological outcomes and showed a trend toward worse survival compared to manual CPR (n = 373). Comparing LDB-CPR to manual CPR, survival to hospital discharge was 5.8% vs 9.9% (p=0.06). The two best cerebral performance categories at hospital discharge were recorded in 3.1% of LDB-CPR patients compared to 7.5% of manual CPR patients (p=0.006). This is a reversal on the use of automated chest compression devices for resuscitation following out-of-hospital cardiac arrest. | 2014. Cochrane review. "Widespread use of mechanical devices for chest compressions during cardiac events is not supported by this review. More RCTs that measure and account for the CPR process in both arms are needed to clarify the potential benefit to be derived from this intervention." (**Brooks et al., 2011**) |

*Table 2 continued on next page*

*Table 2 continued*

| RCT and medical discipline | Reversal summary | Systematic review conclusion |
| --- | --- | --- |
| Harris et al. 2013. Universal Glove and Gown Use and Acquisition of Antibiotic-Resistant Bacteria in the ICU A Randomized Trial. *JAMA* **310**:1571–1580. (10/16/2013) [Critical care] | The emergence of antibiotic-resistant bacteria has become a serious public health issue. To help prevent the spread of these organisms, policies recommending contact precautions (e.g. gloves and gowns) were made by the Centers for Disease Control and Prevention (***Manian and Ponzillo, 2007***). In this trial, intensive care units (ICUs) were randomized to usual care of ICUs (N = 10) or a protocol where all health care workers are required to wear gloves and gowns for all patient contact (intervention ICUs; N = 10). There was no difference in the acquisition of methicillin-resistant *Staphylococcus aureus* or vancomycin-resistant Enterococcus between ICUs that had universal glove and gown use and those that did not (difference, −1.71 acquisitions per 1000 person-days, 95% CI, −6.15 to 2.73; p=0.57). This is a reversal of requiring that all health care workers in ICUs wear gloves and gowns for all patient contact and when entering a patient room. | 2014. 'Contact precautions did not significantly reduce the VRE acquisition rate." (***De Angelis et al., 2014***) This review did not include the RCT. |
| Binanay et al. 2005. Evaluation Study of Congestive Heart Failure and Pulmonary Artery Catheterization Effectiveness: The ESCAPE Trial. *JAMA* **294**:1625–1633. (10/5/2005) [Cardiovascular] | Pulmonary artery catheterization (PAC) was introduced in the 1970 s and was adopted nationwide in the ICU and perioperative settings for congestive heart failure (***Gore et al., 1987***). Although therapies have improved over the years, patients with heart failure still have up to 35–40% one-year mortality rates (***Lee et al., 2003***). PAC has been questioned for its safety and efficacy. This study investigated the survival rate of patients after PAC (n = 206) or clinical assessment alone (n = 207). They found that PAC increased adverse events (21.9% PAC vs 11.5% clinical assessment; p=0.04) and had no effect on days alive out of the hospital during the first 6 months (133 vs 135 days; HR, 1.00; 95% CI, 0.82–1.21; p=0.99), overall mortality (10% vs 9%; OR, 1.26; 95% CI, 0.78–2.03; p=0.35), and number of days hospitalized (8.7 vs 8.3; HR, 1.04; 95% CI, 0.86–1.27; p=0.67). This is a reversal of PAC for patients with congestive heart failure. | 2013. Cochrane review. "PAC is a diagnostic and haemodynamic monitoring tool but not a therapeutic intervention. Our review concluded that use of a PAC did not alter the mortality, general ICU or hospital LOS, or cost for adult patients in intensive care." (***Rajaram et al., 2013***) |
| Schoor et al. 2003. Prevention of Hip Fractures by External Hip Protectors A Randomized Controlled Trial. *JAMA* **289**:1957–1962. (4/16/2003) [Orthopedic] | Hip fractures affect millions of people annually and external hip protectors were designed to absorb the impact of a fall to prevent fractures. There were a number of RCTs investigating external hip protectors and hip fracture prevention showing with mixed results, (***Chan et al., 2000***; ***Lauritzen et al., 1993***; ***Parker et al., 2005***) yet protectors were still regularly prescribed in practices (***van Schoor et al., 2002***). This study found that prescribing a hip protector was not effective in preventing hip fractures in elderly persons aged 70 years and older compared to risk and bone health information. There were 18 fractures in the intervention group (n = 276) compared to 20 fractures in the control group (n = 285; p=0.86). This is a reversal of external hip protectors for preventing fractures in elderly persons in institutional homes. | 2014. Cochrane review. After excluding studies with high risk of bias, this Cochrane systematic review found that hip protectors did not have a significant effect on risk of hip fractures in institutional settings (***Santesso et al., 2014***). |

*Table 2 continued on next page*

*Table 2 continued*

| RCT and medical discipline | Reversal summary | Systematic review conclusion |
|---|---|---|
| Coleman et al. 2012. A Randomized Trial of Nicotine-Replacement Therapy Patches in Pregnancy. *NEJM* **366**:808–818. (3/1/2012) [Obstetrics and gynecology] | Cigarette smoking during pregnancy increases the risks of pregnancy complications, as well as the chance of delivering a low-birth-weight or preterm baby. Despite these risks, approximately 6% to 22% of pregnant women in high-income countries smoke, making cigarette smoking one of the leading causes of adverse pregnancy outcomes (*Cnattingius, 2004*). Behavioral counseling is recommended for pregnant smokers, (*Lumley et al., 2009*) as is nicotine-replacement therapy, which is recommended by several guidelines (*Coleman et al., 2012a*). In the SNAP trial (N = 1050), pregnant smokers receive behavioral counseling and were randomly assigned to either a standard course of nicotine patches or placebo. In this trial, it was found that a nicotine patch was no more effective than placebo in helping pregnant women to quit smoking(9.4% and 7.6%, respectively; unadjusted odds ratio with nicotine-replacement therapy, 1.26; 95% confidence interval, 0.82 to 1.96). This is a reversal on nicotine replacement therapy patches in pregnancy. | 2015. Cochrane review. "NRT [Nicotine Replacement Therapy] used in pregnancy for smoking cessation increases smoking cessation rates measured in late pregnancy by approximately 40%. There is evidence, suggesting that when potentially-biased, non-placebo RCTs are excluded from analyses, NRT is no more effective than placebo. There is no evidence that NRT used for smoking cessation in pregnancy has either positive or negative impacts on birth outcomes (*Coleman et al., 2012b*). |
| Nicolaides et al. 2016. A Randomized Trial of a Cervical Pessary to Prevent Preterm Singleton Birth. *NEJM* **374**:1044–1052. (3/17/2016) [Obstetrics and gynecology] | The transvaginal placement of a silicone pessary is often recommended for pregnant women with a short cervix given their increased risk of spontaneous delivery prior to 34 weeks of gestation. It is believed that this device reduces direct pressure on the cervix and prolongs pregnancy (*Arabin et al., 2003*). This randomized trial compared spontaneous preterm births among women with pessaries with those who underwent expectant management and found that the pessary had no significant effect on the rate of preterm delivery (12.0% and 10.8%, respectively; odds ratio in the pessary group, 1.12; 95% confidence interval, 0.75 to 1.69; p=0.57). This is a reversal on a cervical pessary to prevent preterm singleton birth of women 16 years or older with a cervical length of 25 mm or less. | 2017. "In singleton pregnancies with a [transvaginal ultrasound cervical length] TVU CL $\leq$ 25 mm at 200–246 weeks, the Arabin pessary does not reduce the rate of spontaneous preterm delivery or improve perinatal outcome." *Saccone et al., 2017* |
| Shroyer et al. 2017. Five-Year Outcomes after On-Pump and Off-Pump Coronary-Artery Bypass *NEJM* **377**:623–632. (8/17/2017) [Cardiovascular] | Observational studies in the 1990s showed an association between off-pump coronary-artery bypass and better early clinical outcomes compared to on-pump, and the practice of performing coronary-artery bypass surgery on a beating heart repopularized (*Cleveland et al., 2001*). Yet randomized controlled trials have not been able to show efficacy in off-pump surgeries and suggested that incomplete revascularization was more frequent with off-pump surgery (*Hattler et al., 2012*). This follow-up study (n = 2203) found that 5 year outcomes of death from any cause (relative risk, 1.28; 95% confidence interval [CI], 1.03–1.58; p=0.02) and any major adverse cardiovascular events (relative risk, 1.14; 95% CI, 1.00–1.30; p=0.046) were worse for patients who underwent coronary-artery bypass surgery off-pump compared to on-pump. This is a reversal of off-pump coronary-artery bypass. | 2018. "This meta-analysis represents a comprehensive summary of RCTs comparing OPCABG to ONCABG. Our results showed that OPCABG was associated with no reduction in operative risk, an excess mortality at follow-up $\geq$3 years, and a trend toward higher risk of repeated revascularization.' (*Gaudino et al., 2018*) |

*Table 2 continued on next page*

*Table 2 continued*

| RCT and medical discipline | Reversal summary | Systematic review conclusion |
| --- | --- | --- |
| Friedly et al. 2014. A Randomized Trial of Epidural Glucocorticoid Injections for Spinal Stenosis. *NEJM* **374**:11–21. (7/3/2014) [Neurology] | The treatment of symptomatic lumbar stenosis has included epidural glucocorticoid injections (*Harrast, 2008*). This treatment is frequently prescribed by physicians to treat lumbar stenosis and other conditions, with an estimated 25% of the Medicare population and 74% of patients at the Veteran's Administration being prescribed this treatment. As the usage of glucocorticoid injections increased to treat various ailments, so did the cost. From 1994 to 2001 there was a 271% growth in usage of the treatment, and the cost went from $24 million to over $175 million (*Friedly et al., 2007*). The LESS trial (N = 441) was designed to compare the effectiveness of epidural injections of glucocorticoids plus anesthetic vs. injections of anesthetic alone. At six weeks after randomization, there were no significant differences in RMDQ scores (used to measure functional disability), (adjusted difference in the average treatment effect between the glucocorticoid–lidocaine group and the lidocaine-alone group, −1.0 points; 95% confidence interval [CI], −2.1 to 0.1; p=0.07), or pain intensity, (adjusted difference in the average treatment effect, −0.2 points; 95% CI, −0.8 to 0.4; p=0.48), between the patients treated with glucocorticoids plus lidocaine and those in the lidocaine alone group. This is a reversal of administering epidural glucocorticoid injections in patients who have lumbar central spinal stenosis and moderate-to-severe leg pain and disability. | 2015. AHRQ. "Evidence was limited for epidural corticosteroid injections versus placebo interventions for spinal stenosis (SOE: low to moderate) or nonradicular back pain (SOE: low), but showed no differences in pain, function, or likelihood of surgery." (*Chou, 2015*)*Møller et al., 2012* |
| Lamy et al. 2016. Five-Year Outcomes after Off-Pump or On-Pump Coronary-Artery Bypass Grafting. *NEJM* **375**:2359–2368. (12/15/2016) [Cardiovascular] | Coronary-artery bypass grafting (CABG) can be performed either with a still heart (on-pump CABG) or without a cardiopulmonary bypass on the beating heart (off-pump). Traditionally, surgeons performed surgery on the arrested heart, on-pump CABG, which allowed for increased surgical precision (*Shroyer et al., 2009*). However, surgeons grew concerned that the cross clamping of the aorta, necessary for the on-pump CABG procedure, may be harmful to patients and increase mortality and risk of stroke or other systemic embolic events in these patients. The off-pump method, operating on a beating heart, was developed to decrease the perioperative risks (*Grover, 2012*). However, the clinical literature reported different results about the relative efficacy of off-pump CABG as compared with on-pump CABG (*Lamy et al., 2012*). The CORONARY trial (n = 4752) compared on-pump and off-pump CABG surgery in patients with coronary heart disease. This follow up study on the results of the CORONARY trial found the after 4 years, patients who underwent on-pump and off-pump CABG has similar rates of outcomes of death, stroke, myocardial infarction, renal failure, and repeat revascularization (23.1% % and 23.6%, respectively; hazard ratio with off-pump CABG, 0.98; 95% confidence interval [CI], 0.87–1.10; p=0.72). The costs between the two treatments was similar as well. This is a reversal of off-pump coronary-artery bypass grafting. | 2018. "Off-pump CABG increases long-term (≥5 years) mortality compared with on-pump CABG, based on a meta-analysis of eight medium- to large-size RCTs enrolling a total 8780 patients.' (*Takagi et al., 2017*) 2012. Cochrane review. "Our systematic review did not demonstrate any significant benefit of off-pump compared with on-pump CABG regarding mortality, stroke, or myocardial infarction. In contrast, we observed better long-term survival in the group of patients undergoing on-pump CABG with the use of cardiopulmonary bypass and cardioplegic arrest. Based on the current evidence, on-pump CABG should continue to be the standard surgical treatment. However, off-pump CABG may be acceptable when there are contraindications for cannulation of the aorta and cardiopulmonary bypass. Further randomised clinical trials should address the optimal treatment in such patients." (*Møller et al., 2012*) This review precedes study. |

*Table 2 continued on next page*

*Table 2 continued*

| RCT and medical discipline | Reversal summary | Systematic review conclusion |
| --- | --- | --- |
| Katz et al. 2013. Surgery versus Physical Therapy for a Meniscal Tear and Osteoarthritis. *NEJM* **368**:1675–1684. (5/2/2013) [Orthopedic] | Clinicians who suspect a tear in the meniscus may refer patients either to a surgeon for arthroscopic partial meniscectomy or to physical therapy. This procedure is frequently done in the United States; one estimate is that more than 465,000 patients receive this procedure annually (***Bozic et al., 2012***). Given the frequency and cost of arthroscopic partial meniscectomy and lack of concrete evidence on the clinical benefit of the procedure, the METEOR trial was designed to assess the efficacy of arthroscopic partial meniscectomy surgery as compared with a physical-therapy for increasing physical function of patients with a meniscal tear and moderate osteoarthritis (***Katz et al., 2013***). METEOR found that there was not a significant decrease in the WOMAC score—a measure of physical function in which a higher score means worse physical function—between the patients undergoing surgery and those receiving initial physical therapy. The WOMAC score after 6 months was 20.9 points (95% confidence interval [CI], 17.9 to 23.9) in the surgical group and 18.5 (95% CI, 15.6 to 21.5) in the physical-therapy group (mean difference, 2.4 points; 95% CI, −1.8 to 6.5). The authors conclude that the finding of the METEOR trial advocates for an initial nonoperative strategy for treatment. This is a reversal of surgery for a meniscal tear detected on magnetic resonance imaging (MRI) and osteoarthritis in patients 45 year of age or older. | 2016. "Further evidence is required to determine which patient groups have good outcomes from each intervention. Given the current widespread use of arthroscopic meniscal surgeries, more research is urgently needed to support evidence-based practice in meniscal surgery in order to reduce the numbers of ineffective interventions and support potentially beneficial surgery." (***Monk et al., 2017***) |

DOI: https://doi.org/10.7554/eLife.45183.005

evidence of the use of a newer practice was sometimes easier to find because it had come about during a time when there was more internet use. Conversely, documented evidence of an older practice was sometimes easier to find because there had been more historical commentary about its use. Because of this, newer or more recent practices may be more or less likely to be categorized as established than older or less recent practices. Third, others may categorize results differently, depending on background expertise of the investigators.

To help overcome this limitation, physicians in the clinical setting from a range of backgrounds were invited to review and comment on practices identified as reversals. Our dataset is presented in full in *Supplementary file 2*. It is inevitable that others may feel differently and choose to reclassify some of our examples. We hope our work may serve to enhance and expand upon other efforts to identify and disincentivize low-value practices. Fourth, we relied on the study authors' point of view on whether the results were positive or negative, and there may be reasonable differences of opinion regarding the interpretation of some studies.

Fifth, we did not evaluate the quality of the meta-analysis used to confirm or refute the medical reversal. However, we tried to find the most recent review that was published in either Cochrane or medical journal (for that specialty) to confirm or refute the reversal. Finally, our definition of established may be broad in that we did not limit established practices to only those that were being used widespread, in part because once a practice has been adopted, even intermittently, it is difficult to get patients and patients to abandon this practice. We did, however, maintain that proof of establishment needed to codified into guidelines or be one for which we could prove use outside of a clinical trial or clinical protocol. Additionally, multiple physicians reviewed each practice to confirm that these practices were indeed reversals.

Our primary research objective was to compile a comprehensive review of medical reversals for the benefit of both medical professionals and lay persons. This type of work is fundamentally descriptive and does not seek to test a binary hypothesis. Nevertheless, there are a number of concepts and lessons that may be realized from the results. The breadth of reversals across the

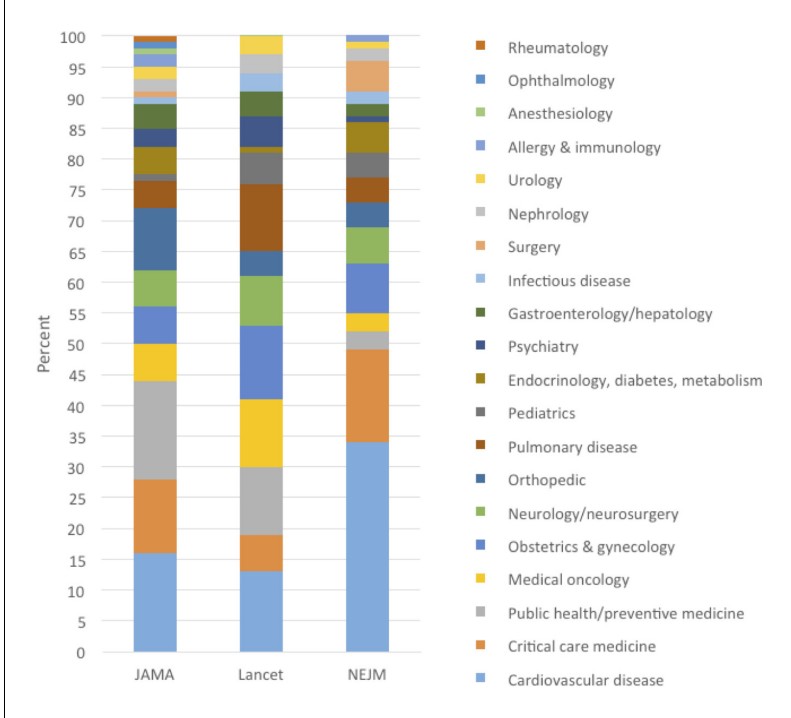

**Figure 2.** Percent of medical reversals in each medical specialty, by journal: JAMA (2003–2017), Lancet (2003–2017), NEJM (2011–2017).
DOI: https://doi.org/10.7554/eLife.45183.004

various fields of medicine emphasize the importance of conducting randomized trials for both novel and established practices. While it may impractical, if not impossible, to test every medical practice in a randomized setting, there are many testable practices that are adopted based on nonrandomized data or bio plausibility. There is a danger in expediting treatments into practice without data proving their efficacy. Once an ineffective practice is established, it is difficult to convince practitioners to abandon its use; eliminating a reversal from standard practice occurs slowly and with resistance (*Prasad et al., 2012*; *Tatsioni et al., 2007*). By aiming to test novel treatments before they are widespread, we can reduce the number of reversals in practice and prevent harms to patients and to the reputation of the medical field. We hope these findings propel medical professionals to critically evaluate their own practices and, going forward, demand high-quality research before adopting a practice, especially for practices that are costlier and/or more aggressive than standard of care.

## Conclusions

We have identified 396 medical reversals spanning different types of medical disciplines, types of interventions, and populations. The de-adoption of these and other low-value medical practices will lead to cost savings and improvements in medical care.

## Methods

### Aim of study

We sought to compile a list of medical reversals that appeared in three leading general medical journals during a 15 year period.

### Search strategy

We used methods similar to our prior survey of 10 years of publications in one high-impact journal (*Prasad et al., 2013*). We reviewed all articles under the headings 'Original Investigation', 'Preliminary Communications', 'Caring for the Critically-Ill Patient', 'Brief Reports', 'Clinical investigations', 'Toward Optimal Laboratory Use', and 'Original Contribution' in JAMA and all articles under the heading 'Articles' in the Lancet from years 2003 to 2017. We reviewed all articles under the heading 'Original Articles' in NEJM from years 2011–2017. The years 2001 to 2010 of the NEJM were previously reviewed and reported (*Prasad et al., 2013*; *Prasad et al., 2011*). The choice of journals was made based on the three general medical journals with the highest 5 year Hirsch index for medical journals (https://jcr.incites.thomsonreuters.com/JCRJournalHomeAction.action). This study was conducted from March 1, 2017 through November 11, 2018.

### Article inclusion

We identified all randomized trials of a clinical practice, or, in other words, any investigation that assessed screening, diagnostic testing, medication(s), procedure(s), surgery, medical device, treatment algorithms, or any change in health care provision systems. We excluded randomized controlled trials (RCTs) that did not concern a medical practice (e.g. a RCT that tested a biological question, such as the effect of testosterone on muscle mass) or that were individual-level patient meta-analyses.

We then excluded trials of novel practices, defined as practices only used in the confines of clinical trials. Established practices were included and defined as those used regularly outside of research trials. This could include off-label use or use outside of the US.

Next, we excluded trials that reached positive or inconclusive results. An article was considered

positive if the trial met its primary endpoint and negative if it failed to meet the primary outcome or if the study measured a hard endpoint (quality of life, mortality, etc.) and failed to show statistical superiority over a prior or lesser standard of practice in the control arm. For non-inferiority or equivalence studies, meeting the pre-specified margin would be considered positive. For studies comparing two established interventions, the more expensive intervention needed to show benefit to be considered positive. Studies were deemed inconclusive if they demonstrated neither clear benefit nor harm (e.g., improved overall survival but no improvement in functional capacity in patients who have had a stroke) or the study was stopped early for reasons other than futility or adverse events.

For each tentative reversal in our dataset, we performed a two-part search to find a systematic review. Meta analyses and/or systematic reviews (MA/SRs) were sought for each RCT designated as a 'reversal' to determine whether the established practice was found to be ineffective across multiple studies. MA/SRs were found by searching, in this order: review articles that cited that trial in Pubmed.gov; review articles that cited the trial in Google Scholar; and then using search terms in Google Scholar. In some cases, MA/SRs were found using the journal website under 'citing articles'. Because of the high-quality review process, Cochrane reviews were first choice for reviews on the article's subject, but if there was no Cochrane review, a meta-analysis from another high-quality journal was used. More recent meta-analyses were prioritized over older meta-analyses on the same topic, and meta-analyses that population-weighted their analyses were prioritized over ones that did not. MA/SRs were categorized as 1) confirming reversal, 2) refuting reversal, 3) insufficient data on reversal, or 4) no MA/SR found. MA/SRs needed to include the RCT in order to be considered as a confirmation of a reversal, and the conclusions needed to be based on results from RCTs only (not on observational or nonrandomized studies). Articles with MA/SRs refuting the reversal were excluded from the final analysis. A table of all confirmed reversals can be found in *Supplementary file 2*.

For all steps of study selection, two reviewers (DH, AH, TC, JG) independently examined information for each article. When there were differences in opinion between the two reviewers, adjudication first involved discussion between the two readers to see whether agreement could be reached. If disagreement persisted, a third reviewer (VP) adjudicated the discrepancy. *Figure 1* shows our study selection strategy.

## Data abstraction and coding

Articles were coded by discipline (public health/general preventive medicine, psychiatry, neurology/neurosurgery, radiation oncology, surgery, urology, allergy and immunology, anesthesiology, dermatology, pediatrics, obstetrics and gynecology, ophthalmology, orthopedic surgery, cardiovascular disease, critical care medicine, endocrinology, diabetes, and metabolism, gastroenterology/hepatology, hematology, infectious disease, medical oncology, nephrology, pulmonary disease, or rheumatology) with the option of a secondary discipline, if it could be categorized as more than one, whether the study was done in a high-income country or a low- to middle- income country (*International Statistical Institute, 2018*), and the type of intervention (medication, procedure, device, screening test, over-the-counter medication, vitamins/supplements/food, behavioral therapy, treatment algorithm, diagnostic instruments, system intervention/quality and performance measure, or optimization). We also abstracted the funding source(s) and categorized the data as industry only, non-industry only, a combination of industry and non-industry sources, or a combination of non-industry and either an insurance company or banking institution. Intervention materials provided by an industry source qualified as having funding support from industry sources.

For all coding, two reviewers (DH, AH, TC, JG) independently extracted information for each article. The aforementioned procedure to resolve disagreement was used.

Four physicians (AC, MH, CL, DM) reviewed all reversals, systematic reviews, and documentation to confirm that the practice was a reversal. Further discrepancies were adjudicated by VP. Thus, our process involved iterative assessment and documentation of practices by a group of researchers and physicians.

## Data analysis

Data are presented using descriptive statistics. Analyses were conducted using Microsoft Excel and R, package Tidyverse (*Wickham, 2017*). This study was not submitted for Institutional Review Board approval because it involved publicly available data and did not involve individual patient data. All abstracted data are included the manuscript and supporting files.

**Diana Herrera-Perez** is in the Knight Cancer Institute, Oregon Health & Science University, Portland, United States

**Alyson Haslam** is in the Knight Cancer Institute, Oregon Health & Science University, Portland, United States

https://orcid.org/0000-0002-7876-3978

**Tyler Crain** is in the Knight Cancer Institute, Oregon Health & Science University, Portland, United States

**Jennifer Gill** is in the Knight Cancer Institute, Oregon Health & Science University, Portland, United States

https://orcid.org/0000-0002-5591-6855

**Catherine Livingston** is in the School of Medicine, Oregon Health & Science University, Portland, United States

**Victoria Kaestner** is in the Knight Cancer Institute, Oregon Health & Science University, Portland, United States

**Michael Hayes** is in the Division of Internal Medicine, Oregon Health & Science University, Portland, United States

**Dan Morgan** is in Department of Epidemiology & Public Health, University of Maryland School of Medicine, Baltimore, United States

**Adam S Cifu** is in Department of Medicine, University of Chicago, Chicago, United States

**Vinay Prasad** is in the Knight Cancer Institute, the Department of Public Health and Preventive Medicine, the Center for Health Care Ethics and the Department of Medicine, Oregon Health & Science University, Portland, United States

prasad@ohsu.edu

*Author contributions:* Diana Herrera-Perez, Conceptualization, Data curation, Formal analysis, Writing—original draft, Writing—review and editing; Alyson Haslam, Conceptualization, Data curation, Formal analysis, Methodology, Writing—original draft, Project administration, Writing—review and editing; Tyler Crain, Conceptualization, Data curation, Formal analysis, Methodology, Writing—original draft, Writing—review and editing; Jennifer Gill, Conceptualization, Data curation, Formal analysis, Methodology, Writing—review and editing; Catherine Livingston, Investigation, Writing—review and editing; Victoria Kaestner, Conceptualization, Data curation, Writing—review and editing; Michael Hayes, Dan Morgan, Adam S Cifu, Writing—review and editing; Vinay Prasad, Conceptualization, Data curation, Formal analysis, Supervision, Funding acquisition, Investigation, Methodology, Writing—original draft, Project administration, Writing—review and editing

*Competing interests:* Adam S Cifu: ASC reports that he receives royalties from the book, Ending Medical Reversal. Vinay Prasad: VP reports that he receives royalties from his book Ending Medical Reversal; that his work is funded by the Laura and John Arnold Foundation; that he has received honoraria for Grand Rounds/lectures from several universities, medical centers, nonprofit groups and professional societies; that he is a writer for Medscape; and that he hosts the podcast Plenary Session (which has Patreon Backers). The other authors declare that no competing interests exist.

### Funding

| Funder | Author |
| --- | --- |
| Laura and John Arnold Foundation | Vinay Prasad |

The funders had no role in study design, data collection and interpretation, or the decision to submit the work for publication.

**Decision letter and Author response**
Decision letter https://doi.org/10.7554/eLife.45183.011
Author response https://doi.org/10.7554/eLife.45183.012

## Additional files
### Supplementary files
• Supplementary file 1. Funding sources for articles in which a medical reversal was identified.
DOI: https://doi.org/10.7554/eLife.45183.006
• Supplementary file 2. Reversal summaries identified, by journal.
DOI: https://doi.org/10.7554/eLife.45183.007
• Transparent reporting form
DOI: https://doi.org/10.7554/eLife.45183.008

### Data availability
Data were obtained from publicly available data and are included in the manuscript and supporting files.

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
