## [Decision Letter]

Thank you for submitting your article "A Comprehensive Analysis of Recent Medical Reversals that includes 396 Low-Value Practices in the Biomedical Literature" for consideration by *eLife*. Your article has been reviewed by three peer reviewers, and the evaluation has been overseen by Eduardo Franco acting as Reviewing Editor and Senior Editor.

As standard practice in *eLife*, the reviewers have discussed their critiques with one another in an online panel moderated by the Reviewing Editor. After consensus was reached, the Reviewing Editor has drafted this decision to help you prepare a revised submission.

Summary:

The authors aimed to identify low-value care or medical reversals from randomized controlled trials (RCT) published in three top-tier medical journals from 2003 to 2017. RCTs were included if they were on an established practice, and then whether the trial had negative results. The authors classified the type of intervention and discipline, and searched for a systematic review on the practice. Identification of medical reversals across disciplines is an enormous task and the authors have done well to carry this out systematically, and created a valuable resource (the list of 396 medical services) for de-adoption efforts. This paper partly overlaps with, but greatly extends the group's previous work building a sample of medical reversals.

Essential revisions:

Main issue: All reviewers raised concerns about lack of clarity on the aims of your work. Was your goal to identify practices that could be targeted for de-implementation? If that is the main purpose, this could be stated more clearly and the Discussion could better explain next steps. For example, would there be an additional layer of review before de-implementation campaigns? Are there additional questions that would need to be addressed to advance some of these treatments into a concerted de-implementation plan? The manuscript ended a bit abruptly and there might be a missed opportunity to discuss issues on evidence and whether/how these "396 medical reversals" indicate a problem.

1) The title could be more specific in referring to JAMA, Lancet, NEJM. It is a comprehensive analysis of those three journals, not 'the biomedical literature' which of course is far broader. The appropriate title can be discussed with the Features Editor.

2) General: Why was the BMJ not included as the fourth highest general medical journal? The BMJ has a strong track record of publishing work that aligns with the reversal agenda. Also, NEJM and JAMA tend to be very US-centric but both Lancet and BMJ (albeit UK based) tend to me more outward looking to the rest of the world. Given the degree of publication bias that works against reversals coming to light, the BMJ might offer high marginal yield and the fact that it was excluded may have been a missed opportunity. Please elaborate in more detail in the Discussion the choice of journals.

3) General: Is it possible that the definition of reversals was overly permissive? What might be more compelling is if the authors can document that interventions had been recommended in clinical practice guidelines at some point before the trial result. Or at least- it would be nice if evidence were presented that the intervention was in widespread use. It is possible that some interventions that "reversed" were not actually in widespread use and/or recommended.

4) Abstract: "This may serve as a starting point for de-adoption efforts and the study of the utilization of these practices." This may be one starting point but with the proliferation of other lists (e.g., Choosing Wisely [CW]) this is a parallel or adjunct starting point. While this is likely a more robust starting point, the reality is CW lists are front and center right now so are attracting more attention than they probably warrant from an evidentiary basis.

5) Introduction: Readers would benefit from a definition of reversal as used in this study, and in particular if the focus is on interventions in toto, or their application by/to patient subgroups (e.g., accounting for heterogeneity of treatment effect), or both. This might be gleaned from Table 2 but it needs addressing upfront in main text (probably Introduction).

6) Introduction: "…and identifying these practices is a starting point in for reducing cost and improving care." Again, with this framing you are missing an opportunity for this paper. There are so many lists of LVC now that identification is less of a challenge, prioritization from the lists perhaps more so, particularly for policymakers. The robust method you have undertaken and results offer as much if not more of a prioritization opportunity, than an identification opportunity. In short, taken in isolation it is identification, but viewed through the bigger picture it offers a prioritization angle, particularly given CW approaches by individual societies have used less formal methods.

7) Introduction, paragraph two: This paragraph is a bit confusing. It may be difficult to name (or identify?) these practices for many reasons beyond issues with the scope of the Cochrane review database. This should be expanded on.

8) Introduction, paragraph three:. This paragraph hints at some of the good reasons to go 'beyond' the CW lists to identify medical reversals, but these probably aren't obvious to readers unfamiliar with the campaign and I think really important to justify why a broader search strategy is necessary to identify low-value practices.

9) Introduction, paragraphs five and six: Referencing and focusing on the earlier study on the identified NEJM RCTs would make more sense here, rather than focus so much on the Schwartz 2014 study since the current article has very different aims/results compared to this paper.

10) Methods: For the second step (search out of systematic reviews)- how did the authors handle quality of SR/MA as well as instances where SR/MA's came to conflicting conclusions?

11) Results: The aim of the article was to compile a list of medical reversals. This is a challenge for the Results section, because there are many possible descriptive statistics from this list that the authors could choose to present (Table 1) – none of which is specific to the aim of the article. The Results (and/or Methods) section could improve with justification of why specific results were included.

12) Results: the paper does not appear to have been driven by a particular hypothesis- which makes it feel more like a gallery or reference work than a research report. It would have helped to give context if hypotheses were articulated and perhaps some relationships tested. For example, one wondered: who funded the "reversal" studies? How large were the reversals – did they tend to barely pull SR estimates over the null, or did they exert decisive effects?

Related to the above- it is not immediately clear to the reader what the message is here. Clearly, it is sometimes important to run randomized trials, since there can be residual uncertainty about efficacy. But there is no point in running trials if something is already proven. So the manuscript leaves lots of questions unanswered- are too many treatments reversed? Is there not enough research aimed at reversals? Does too much time pass between uptake and reversal? What types of interventions are at greater risk of an extended period of "reversal latency?" A better concluding paragraph would help pull the piece together. Right now it sort of ends abruptly.

13) Table 2 (and also the supplementary file 2 with all reversals) would be improved if they clearly stated the authors' classification of the medical discipline and intervention type. The authors stated this as a potential limitation and potential disagreement/debate with readers in the Discussion, so including these in Table 2 would at least give some insight into the authors' logic.

14) The authors also state that they hope the full list of reversals would be a resource for "de-adoption efforts", so why not include the medical discipline in the supplementary file. This seems like an easy step to make it much more useful to specialists and policy-makers interested in disinvestment in their area.

15) It is also not clear how the supplementary file is organised. Why is there a split between the tables and references after intervention 57, 81 and so on? It would be helpful to have some headings or descriptions here. PubMed IDs (PMID) should be added to every entry for ready reference.

16) Discussion, paragraph three: Makes a really good point about the patient 'accessed' low-value care/reversals, which hasn't been discussed much in the literature before. Adding something about the need for direction/further research here would be useful to highlight this point more (such as how the public accesses or could be more aware of this evidence).

17) Strengths and limitations: "Studies were deemed inconclusive if they demonstrated neither clear benefit nor harm." The qualifier 'inconclusive' may not be accounting for poor cost-effectiveness (e.g., inconclusive but at a cost, possibly more than a comparator). And harm is poorly measured in RCTs, certainly in relation to the trial evaluation and definitely in relation to its real world use, i.e., harms are incomprehensively measured, particularly those downstream harms (and costs). An 'inconclusive' label should place the burden of proof back on the intervention champions/sponsors to prove safety and effectiveness, rather than let it be perceived as benign. Also the Methods section does not speak to trials that involved measures of equivalence or non-inferiority. If an intervention was found clinically equivalent/non-inferior did it evade further scrutiny in this study? An obvious concern here is marginal cost-effectiveness (i.e., EQ/NI but more expensive) or questions about sufficient measuring and reporting of harm. Please clarify.

18) "…and the conclusions needed to be based on results from RCTs only (not on observational or nonrandomized studies). Articles with MA/SRs refuting the reversal were excluded from the final analysis." But all RCTs are not created equal. Did you carry out quality appraisal and critique? Were RCTs with higher degrees of design rigour (adequate blinding, sham arms, use of appropriate comparator/s etc) offered more weight than those of less rigour? If not then might this mean your methods are quite conservative and therefore specific? (i.e., some of your excluded studies might be false negatives?) The last sentence in 'limitations' indicates no quality appraisal was conducted. If so this heightens the risk of false negatives, and this would be a major limitation of the study.

---

## [Author Response]

Essential revisions:Main issue: All reviewers raised concerns about lack of clarity on the aims of your work. Was your goal to identify practices that could be targeted for de-implementation? If that is the main purpose, this could be stated more clearly and the Discussion could better explain next steps. For example, would there be an additional layer of review before de-implementation campaigns? Are there additional questions that would need to be addressed to advance some of these treatments into a concerted de-implementation plan? The manuscript ended a bit abruptly and there might be a missed opportunity to discuss issues on evidence and whether/how these "396 medical reversals" indicate a problem.

Our main goal is to identify practices so that they can assist in de-implementation efforts. We have added this statement to the Introduction stating our goal. We have also added to the Conclusion by expanding on why the identification of these 396 medical practices is important.

1) The title could be more specific in referring to JAMA, Lancet, NEJM. It is a comprehensive analysis of those three journals, not 'the biomedical literature' which of course is far broader. The appropriate title can be discussed with the Features Editor.

We have rephrased the wording in the title as has been suggested by the editorial team at *eLife* but are happy to consider others.

2) General: Why was the BMJ not included as the fourth highest general medical journal? The BMJ has a strong track record of publishing work that aligns with the reversal agenda. Also, NEJM and JAMA tend to be very US-centric but both Lancet and BMJ (albeit UK based) tend to me more outward looking to the rest of the world. Given the degree of publication bias that works against reversals coming to light, the BMJ might offer high marginal yield and the fact that it was excluded may have been a missed opportunity. Please elaborate in more detail in the Discussion the choice of journals.

We thank the reviewer for their suggestion and acknowledge that a review of BMJ would have been within the scope of our review. The top three medical journals (NEJM, Lancet, and JAMA) were decided upon in *a priori* discussion with the funder. As such, the funding for this specific project covered time and personnel to review these journals only. This amounted to hundreds of thousands of dollars of expenditure and 7000 person-hours.

3) General: Is it possible that the definition of reversals was overly permissive? What might be more compelling is if the authors can document that interventions had been recommended in clinical practice guidelines at some point before the trial result. Or at least- it would be nice if evidence were presented that the intervention was in widespread use. It is possible that some interventions that "reversed" were not actually in widespread use and/or recommended.

This is a fair point. While there is the possibility that our coding of established is overly permissive, we attempted to minimize this. We did so by defining an ‘existing practice’ to be one that is codified into practice by guidelines, or one for which we could prove use outside of a clinical trial or clinical protocol. Of course, this definition is broad, and the degree to which a practice is used may range from some use to near total use. The solution to this would be to perform an administrative database search for every practice we considered. While this was not in the scope of the current work, it will be done for selected reversals in years to come. We have added this as a limitation to our study.

4) Abstract: "This may serve as a starting point for de-adoption efforts and the study of the utilization of these practices." This may be one starting point but with the proliferation of other lists (e.g., Choosing Wisely [CW]) this is a parallel or adjunct starting point. While this is likely a more robust starting point, the reality is CW lists are front and center right now so are attracting more attention than they probably warrant from an evidentiary basis.

You are correct to note this is an adjunct starting point. We have reworded the conclusion here to indicate that these results enhance and expand upon previous efforts (in the Abstract and in the conclusion).

5) Introduction: Readers would benefit from a definition of reversal as used in this study, and in particular if the focus is on interventions in toto, or their application by/to patient subgroups (e.g., accounting for heterogeneity of treatment effect), or both. This might be gleaned from Table 2 but it needs addressing upfront in main text (probably Introduction).

Thank you. Great point. We have clarified the definition for medical reversal in the Introduction.w

6) Introduction: "…and identifying these practices is a starting point in for reducing cost and improving care." Again, with this framing you are missing an opportunity for this paper. There are so many lists of LVC now that identification is less of a challenge, prioritization from the lists perhaps more so, particularly for policymakers. The robust method you have undertaken and results offer as much if not more of a prioritization opportunity, than an identification opportunity. In short, taken in isolation it is identification, but viewed through the bigger picture it offers a prioritization angle, particularly given CW approaches by individual societies have used less formal methods.

We have added to the Introduction that our broad method may permit others to prioritize which efforts of de-implementation. These practices may concern higher cost or more invasive practices, which may arguably be tackled first.

7) Introduction, paragraph two: This paragraph is a bit confusing. It may be difficult to name (or identify?) these practices for many reasons beyond issues with the scope of the Cochrane review database. This should be expanded on.

We have added several sentences and reworded some to better describe the pros and cons of other methods for identifying low-value practices.

8) Introduction, paragraph three:. This paragraph hints at some of the good reasons to go 'beyond' the CW lists to identify medical reversals, but these probably aren't obvious to readers unfamiliar with the campaign and I think really important to justify why a broader search strategy is necessary to identify low-value practices.

We have added an additional sentence to describe why we conducted this study in the way that we did.

9) Introduction, paragraphs five and six: Referencing and focusing on the earlier study on the identified NEJM RCTs would make more sense here, rather than focus so much on the Schwartz 2014 study since the current article has very different aims/results compared to this paper.

The reviewer is correct in that our study has different aims than the Schwartz 2014 study. Our intent was to highlight the financial ramifications of low-value medical practices. Since the Schwartz paper only looked at the 26 most common practices, the financial outcomes of 396 practices could be even greater. However, because a financial analysis was out of the scope of our study and we were not able to calculate the total costs, we discussed the work by Schwartz and colleagues. We did augment this part of the Discussion by bringing in prior medical reversal findings by Prasad and colleagues (2013).

10) Methods: For the second step (search out of systematic reviews)- how did the authors handle quality of SR/MA as well as instances where SR/MA's came to conflicting conclusions?

We did not do a formal analysis of the quality of the systematic reviews. Our preference was to use Cochrane Reviews because they are recognized for their high-quality methods in conducting systematic reviews and meta-analyses. Of the reversals that were confirmed by a review, 147 (70%) were confirmed by a Cochrane review. In cases where there was no Cochrane review and there were multiple reviews, we used ones that were newer and ones that population weighted their analysis. More detail has been added to the Methods section.

11) Results: The aim of the article was to compile a list of medical reversals. This is a challenge for the Results section, because there are many possible descriptive statistics from this list that the authors could choose to present (Table 1) – none of which is specific to the aim of the article. The Results (and/or Methods) section could improve with justification of why specific results were included.

We are picking a few ways to describe this data, but to be clear this is primarily a descriptive and not hypothesis driven work. We plan to publish our supplement on a website that will permit users to sort the reversals as they wish, and include a search engine. We are working with a web-developer to make this happen.

12) Results: the paper does not appear to have been driven by a particular hypothesis- which makes it feel more like a gallery or reference work than a research report. It would have helped to give context if hypotheses were articulated and perhaps some relationships tested. For example, one wondered: who funded the "reversal" studies? How large were the reversals – did they tend to barely pull SR estimates over the null, or did they exert decisive effects?

This work is primarily non-hypothesis driven, but descriptive. We appreciate the suggestion of adding the funding source to the reported data. We have added these data to the Results section, with additional detail given in the Methods.

Related to the above- it is not immediately clear to the reader what the message is here. Clearly, it is sometimes important to run randomized trials, since there can be residual uncertainty about efficacy. But there is no point in running trials if something is already proven. So the manuscript leaves lots of questions unanswered- are too many treatments reversed? Is there not enough research aimed at reversals? Does too much time pass between uptake and reversal? What types of interventions are at greater risk of an extended period of "reversal latency?" A better concluding paragraph would help pull the piece together. Right now it sort of ends abruptly.13) Table 2 (and also the supplementary file 2 with all reversals) would be improved if they clearly stated the authors' classification of the medical discipline and intervention type. The authors stated this as a potential limitation and potential disagreement/debate with readers in the Discussion, so including these in Table 2 would at least give some insight into the authors' logic.

We have added a column to the Table 2 of the primary medical discipline for each medical practice reversal.

14) The authors also state that they hope the full list of reversals would be a resource for "de-adoption efforts", so why not include the medical discipline in the supplementary file. This seems like an easy step to make it much more useful to specialists and policy-makers interested in disinvestment in their area.

We have added a column that specifies the primary medical discipline for each medical reversal. We will also be including the results of our analysis on a medical reversal website, which we are currently designing. A link to this website will be shared with *eLife*.

15) It is also not clear how the supplementary file is organised. Why is there a split between the tables and references after intervention 57, 81 and so on? It would be helpful to have some headings or descriptions here. PubMed IDs (PMID) should be added to every entry for ready reference.

The supplemental file is arranged in reverse chronological order, by journal. As for the breaks in the table(s), this was done for purely practical reasons – we started a new table when Endnote (the reference software we used) was about to crash. Access to a more powerful computer with specialized RAM capability would solve this issue by allowing us to create a single table with one set of references. The website will have links to all articles that we reference in our write-ups.

16) Discussion, paragraph three: Makes a really good point about the patient 'accessed' low-value care/reversals, which hasn't been discussed much in the literature before. Adding something about the need for direction/further research here would be useful to highlight this point more (such as how the public accesses or could be more aware of this evidence).

We thank the reviewer for this comment and have added a sentence on the need for discussion regarding these interventions and their use in everyday practice.

17) Strengths and limitations: "Studies were deemed inconclusive if they demonstrated neither clear benefit nor harm." The qualifier 'inconclusive' may not be accounting for poor cost-effectiveness (e.g., inconclusive but at a cost, possibly more than a comparator). And harm is poorly measured in RCTs, certainly in relation to the trial evaluation and definitely in relation to its real world use, i.e., harms are incomprehensively measured, particularly those downstream harms (and costs). An 'inconclusive' label should place the burden of proof back on the intervention champions/sponsors to prove safety and effectiveness, rather than let it be perceived as benign. Also the Methods section does not speak to trials that involved measures of equivalence or non-inferiority. If an intervention was found clinically equivalent/non-inferior did it evade further scrutiny in this study? An obvious concern here is marginal cost-effectiveness (i.e., EQ/NI but more expensive) or questions about sufficient measuring and reporting of harm. Please clarify.

These are excellent points by the reviewer. Inconclusive does consider cost-effectiveness as far as the practice works or does not work, although our coding did not explicitly use cost to judge these practices. Reversals are things that did not improve important outcomes, and so we were most interested in the effectiveness of an intervention instead of the cost value of it. Some entities did not make our list if they offered real benefits even tremendous marginal cost. In other words, there may be non-reversals that are not cost effective. However, in many of the studies where there were notable cost differentials between the two treatments, the more expensive intervention was also the novel intervention. There were a few studies that compared two established interventions, where one was the more expensive of the two. In this case, the newer, albeit established, and more expensive would have had to have shown greater benefit for the results to be considered positive. Also, we took our results and took them in the context in which the study was framed. If the study was a noninferior or equivalence study and the intervention met the noninferiority margin, it was considered positive. We have added a few sentences to the Methods section for clarification. As for measuring harm, we did find that some interventions were harmful in a statistically significant way, these studies were coded as negative because they did not meet their primary endpoint. The studies that were coded as inconclusive were ones that may have met their primary endpoint, but other endpoints were as or more meaningful to the patient as the primary endpoint. Such is the case with alteplase for survivors of stroke, where it did prolong overall survival but did not improve functional scores. In this case, there is debate about which outcome is the better one on which to make a conclusive decision about benefits.

18) "…and the conclusions needed to be based on results from RCTs only (not on observational or nonrandomized studies). Articles with MA/SRs refuting the reversal were excluded from the final analysis." But all RCTs are not created equal. Did you carry out quality appraisal and critique? Were RCTs with higher degrees of design rigour (adequate blinding, sham arms, use of appropriate comparator/s etc) offered more weight than those of less rigour? If not then might this mean your methods are quite conservative and therefore specific? (i.e., some of your excluded studies might be false negatives?) The last sentence in 'limitations' indicates no quality appraisal was conducted. If so this heightens the risk of false negatives, and this would be a major limitation of the study.

We have added more detail of how the MA/SR were selected. We did not perform a formal evaluation of the quality of SR/MA. Our preference was to use a Cochrane Review because the quality of their reviews is well established, and in most cases, we were able to do so. In a number of instances, there was only one SR/MA to confirm or refute the reversal. In instances where there were multiple reviews, our methods were to use a Cochrane review first, then another review from a high-quality journal, with newer reviews having priority over older reviews, and reviews that population-weighted their analysis over ones that did not. As for the quality of the individual RCTs, we deferred to the SR/MA to report that.